# Creating Interprofessional Readiness to Advance Age-Friendly U.S. Healthcare

**DOI:** 10.3390/ijerph19095258

**Published:** 2022-04-26

**Authors:** Leland Waters, Sarah A. Marrs, Catherine J. Tompkins, Robert Fix, Sheryl Finucane, Constance L. Coogle, Kevin Grunden, Emily S. Ihara, Madeline McIntyre, Pamela Parsons, Patricia Slattum

**Affiliations:** 1Virginia Center on Aging, College of Health Professions, Virginia Commonwealth University, P.O. Box 980229, Richmond, VA 23298-0229, USA; marrssa@vcu.edu (S.A.M.); ccoogle@vcu.edu (C.L.C.); kgrunden@myinnovage.com (K.G.); mcintyremm@vcu.edu (M.M.); pwslattu@vcu.edu (P.S.); 2Department of Social Work, College of Health and Human Services, George Mason University, 4400 University Drive, MS 1F8, Fairfax, VA 22030, USA; ctompkin@gmu.edu (C.J.T.); eihara@gmu.edu (E.S.I.); 3Department of Occupational Therapy, College of Health Professions, Virginia Commonwealth University, P.O. Box 980008, Richmond, VA 23298-0008, USA; robert.fix@vcuhealth.org; 4Department of Physical Therapy, College of Health Professions, Virginia Commonwealth University, P.O. Box 980224, Richmond, VA 23298-0224, USA; sfinucan@vcu.edu; 5School of Nursing, Virginia Commonwealth University, P.O. Box 980567, Richmond, VA 23298-0567, USA; pparsons@vcu.edu

**Keywords:** older adult, wellness, health promotion, age-friendly practice, what matters, medication, mentation, mobility, health disparities

## Abstract

A successful interprofessional faculty development program was transformed into a more clinically focused professional development opportunity for both faculty and clinicians. Discipline-specific geriatric competencies and the Interprofessional Education Collaborative (IPEC) competencies were aligned to the 4Ms framework. The goal of the resulting program, Creating Interprofessional Readiness for Complex and Aging Adults (CIRCAA), was to advance an age-friendly practice using evidence-based strategies to support wellness and improve health outcomes while also addressing the social determinants of health (SDOH). An interprofessional team employed a multidimensional approach to create age-friendly, person-centered practitioners. In this mixed methods study, questionnaires were disseminated and focus groups were conducted with two cohorts of CIRCAA scholars to determine their ability to incorporate learned evidence-based strategies into their own practice environments. Themes and patterns were identified among transcribed interview recordings. Multiple coders were used to identify themes and patterns and inter-coder reliability was assessed. The findings indicate that participants successfully incorporated age-friendly principles and best practices into their own work environments and escaped the silos of their disciplines through the implementation of their capstone projects. Quantitative data supported qualitative themes and revealed gains in knowledge of critical components of age-friendly healthcare and perceptions of interprofessional collaborative care. These results are discussed within a new conceptual framework for studying the multidimensional complexity of what it means to be age-friendly. Our findings suggest that programs such as CIRCAA have the potential to improve older adults’ health by addressing SDOH, advancing age-friendly and patient-centered care, and promoting an interprofessional model of evidence-based practice.

## 1. Introduction

### 1.1. Rationale

The Age-Friendly Health Systems Initiative, born out of a partnership between the John A. Hartford Foundation, the Institute for Healthcare Improvement, the American Hospital Association, and the Catholic Health Association of the United States, was created as a way to advance the practice of providing age-friendly care [1]. This initiative was created to meet the challenges provided by a growing population of older adults whose care needs are diverse and, sometimes, complex. Age-friendly care possesses three primary qualities: (1) it follows a set of evidence-based practices known as the 4Ms [1] (what matters most, medication, mentation, and mobility); (2) it causes no harm; and (3) it places what matters to the patient and their families/caregivers at the center of care plans. The 4Ms framework advances these ideas by putting the patients’ desires and goals of care at the center of care plan development while also considering the patients’ mobility, medications, and mentation. Incorporated within this framework is an understanding of the necessity of integrating the social determinants of health (SDOH) within the assessment and delivery of care. A critical element to advancing these ideals and practices into the workforce is professional development for faculty and clinicians who provide care to older patients and prepare future cohorts of the healthcare workforce. As such, we sought to develop a faculty and clinician professional development program that would help to advance age-friendly care. Our purpose here is to detail the creation of this program, known as CIRCAA, and share insights from those who have completed the program.

CIRCAA, which stands for Creating Interprofessional Readiness for Complex and Aging Adults, is a faculty and clinician development program for healthcare professionals. The program was developed by an interprofessional team of faculty and clinicians who had appointments at or affiliations with an urban research university in the mid-Atlantic region of the United States. This interprofessional group meets twice a month to oversee a variety of interprofessional geriatrics training initiatives, including CIRCAA. The CIRCAA curriculum was based on the Faculty Development Program (FDP), an interprofessional curriculum grounded in evidence-based practices for healthcare professionals with a faculty appointment. The program was based on a model offered by the University of California, San Francisco, and the curriculum was guided by the Partnership for Health in Aging Workgroup on Multidisciplinary Competencies in Geriatrics [2,3,4,5]. A 2019 report from the Advisory Committee on Interdisciplinary Community-Based Linkages [6] pointed out that current accreditation standards do not adequately address age-friendly concepts and the 4Ms [7]. One recommendation from the report was that health professions programs should integrate age-friendly interprofessional principles into their curricula and should be designed to transform curricular expectations within continuing education programs. We adapted our previous faculty development program into the more clinically focused CIRCAA program to advance age-friendly practice. Taking a multidimensionality approach whereby individual health is determined by several dimensions (i.e., the 4Ms), we sought to create age-friendly practitioners who are knowledgeable of the age-friendly framework and person-centered care.

### 1.2. Program Development

To create the CIRCAA curriculum, the interprofessional faculty group first gathered geriatrics competencies from each of their respective fields and professional organizations. These competencies as well as the Interprofessional Education Collaboration’s (IPEC) [8,9] competencies for interprofessional teamwork were then mapped to the 4Ms and 11 other topics that were identified from the FDP curriculum by the interprofessional faculty team that was informed by feedback from former FDP scholars (Appendix A, Alternate Mapping Competencies). Specifically, discipline-specific geriatric competencies (medicine (medical students, residents, and geriatric psychiatry), nursing (bachelors of science in nursing (BSN) and nurse practitioners (NP)), occupational therapy, chaplaincy, pharmacy, and physical therapy) and Interprofessional Education Collaborative competencies were aligned to the 4Ms framework (what matters most, medication, mentation, and mobility). A focus on interprofessional education was maintained in the transition from FDP to CIRCAA, as interprofessional care supports age-friendly practice. Sub-topics or themes were identified among the grouped competencies to then create individual session foci. Using the session foci and the associated competencies, learning objectives were created for each session, ensuring that the participants would receive information that reflected the competencies associated with that session. The objectives then guided the design of curricular content for each session. Within the curriculum were practicum experiences in age-friendly practices that allowed scholars to observe some of the curriculum content being applied in the field. A capstone project was included within the CIRCAA program with the expectation that scholars would share information related to the 4Ms framework and interprofessional practice with their colleagues or students or apply these principles in their practice setting in the form of action research.

### 1.3. The CIRCAA Curriculum

The first year, CIRCAA was offered in person until March, 2020, when the COVID-19 pandemic forced the program to pivot to virtual education. The second year, the program was offered entirely virtually, with each session shortened from five hours to three, with additional materials made available for self-studying. The capstone project was sustained during the pandemic, but in-person practicum experiences were curtailed. The curriculum outline is in Appendix B: CIRCAA Program Outline and Changes Due to COVID. Participants were recruited through emailed announcements and word of mouth and applied to join the program.

The conceptual themes surrounding CIRCAA training included incorporating health disparities and inequities in care delivery and the role of SDOH and their impact on healthy community living. The foundational FDP maintained an academic didactic format of training focused on preparing interprofessional health profession faculties to incorporate geriatrics into their clinical or classroom teaching. The transition to the CIRCAA format created opportunities to provide evidence-based strategies for translation to practice and then to evaluate the learners’ ability to incorporate this training into their own practice settings. Using the 4M’s framework as a guide, the program is able to focus on the concepts that are most important to improving care delivery and outcomes for older adults across practice settings. The onset of the COVID-19 pandemic required a rapid conversion to a hybrid format in order to sustain the program. Converting to a hybrid format presented initial challenges but also provided an opportunity to recruit from a broader geographic region. In prior years, participants were required to attend in person and to live within driving distance for the on-site monthly sessions. With the conversion to a virtual format, participants were able to join the program from as far away as California. The virtual format also required a transition in the types of learning strategies to be more interactive during group learning sessions with less intensive time commitments while still maintaining the integrity of the program and accomplishing the established goals and objectives. Teaching strategies were adapted and revised over the two years of the COVID-19 pandemic using the Plan Do Study Act (PDSA) method of evaluation and continual refinement, including increased small group learning activities and case studies [10,11,12].

Educational strategies, practice innovation, and evaluation methods were incorporated into the training to help scholars develop their capstone projects, a curriculum enhancement or practice intervention that addresses one of the 4Ms and fills a need in their workplace. Capstone projects could be either a practicum/practice intervention or curriculum-based. CIRCAA scholars implemented practice-based projects to address identified needs in their work settings. Each project was designed to advance at least one of the 4Ms in an interprofessional manner. Many of the projects concerned the 4Ms broadly, while several specifically focused on medication or mentation. Project topics included improving communication for diabetes self-management, de-prescribing in home health, and ageism in health profession training.

## 2. Methods

As CIRCAA was developed, program assessment activities to formatively and summatively evaluate the program were also planned. These activities included an end-of-program focus group with scholars in addition to a pre-program and post-program evaluation questionnaire. The 2020 cohort consisted of 11 scholars, 8 of whom fully completed the program; the remaining 3 scholars partially completed the program and had to reduce engagement due to work constraints caused by the COVID-19 pandemic. The 2020 cohort of scholars represented the following disciplines: nursing (*n* = 3), nurse practitioner (*n* = 1), occupational therapy (*n* = 1), oral health (*n* = 1), pharmacy (*n* = 2), physical therapy (*n* = 2), and public health (*n* = 1). Eleven scholars completed the CIRCAA program as part of the 2021 cohort and represented the following disciplines: gerontology (*n* = 1), health care administration (*n* = 1), medicine (*n* = 1), nursing (*n* = 3), occupational therapy (*n* = 2), physical therapy (*n* = 1), physician assistant (*n* = 1), and social work (*n* = 1).

### 2.1. Quantitative Methods

The pre- and post-program evaluation assessments included a 16-item questionnaire based directly on the IPEC competencies [13,14,15]. The IPEC questionnaire is made up of two subscales: Interprofessional Interactions and Interprofessional Values. Participants respond to items using a 5-point Likert scale ranging from strongly disagree (1) to strongly agree (5). We also asked scholars a series of knowledge questions specifically related to interprofessional practice when caring for older adults and the 4Ms framework (see Table 1). In total, nine 2020 scholars completed both the pre- and post-program questionnaires, and all eleven 2021 scholars completed both the pre-and post-program questionnaires. To compare changes in the IPEC scores, the pre- and post-program domain scores for each subscale were compared using dependent samples *t*-tests including both cohorts.

### 2.2. Qualitative Methods

To understand the experiences and satisfaction level of the CIRCAA participants, two separate focus groups were implemented. The participants included faculty and clinicians from various professions. In 2020, six participants who had experienced CIRCAA via a hybrid model took part in a focus group, and, in 2021, eight focus group participants experienced CIRCAA completely virtually.

The focus groups lasted for about one hour and were recorded and transcribed verbatim. The focus group questions are provided in Appendix C. Three researchers participated in the data analysis utilizing the grounded theory techniques of open coding, memoing, and constant comparative analysis [16]. The transcripts were coded independently, and the researchers met frequently to discuss and agree on codes and themes. Prior to coding, the researchers discussed the grounded theory process of asking the following questions of the data:What are the data a study of?What is actually happening in the data?

Asking these questions is a grounded theory technique that helps to keep the coding conceptual instead of descriptive. A total of 61 different codes emerged from the data during open coding. As a part of the memoing process, one researcher utilized the focus group data to define the codes. At the completion of this process and as a result of constant comparative analysis, two main categories were described.

## 3. Results

### 3.1. Quantitative Results

The average rating on the Interprofessional Interaction subscale before CIRCAA was 4.03 (SD = 0.73), and ratings increased to 4.47 (SD = 0.52) after CIRCAA concluded. Similarly, the average rating on the Interprofessional Values subscale was 4.40 (SD = 0.54), but ratings increased to 4.67 (SD = 0.49) after CIRCAA concluded. Notably, the evaluation data from the pre and post program questionnaires revealed a significant increase in both the interprofessional interactions (t(19) = 3.40, *p* < 0.01, d = 0.76) and interprofessional values (t(19) = 3.22, *p* < 0.01, d = 0.72) subscale scores. Additionally, as can be seen in Table 1, the majority of scholars reported that they felt more knowledgeable of the 4Ms framework, age-friendly health systems, person-centered care, and the roles of other disciplines when caring for older adults.

### 3.2. Qualitative Results

Two main categories of themes emerged from the analysis of the focus group data. The first category, creating age-friendly readiness, included four themes: modeling, innovating, leveraging, and challenging. The second category, understanding and implementing an interprofessional age-friendly framework, established two themes: impacting and cultivating. Each of the categories and their accompanying themes are described in detail below.

### 3.3. Creating Age-Friendly Readiness

One core question that was asked during the focus group sessions that concentrated on participants discussed the strengths and challenges of being trained as age-friendly professionals working within an interprofessional environment. The themes emerging from this discussion were modeling, innovating, leveraging, and challenging.

#### 3.3.1. Modeling

The scholars described the clinical practice and teachings of the interprofessional team they had during their year-long time as scholars as exhibiting the professionalism that they want to model in their own practices and teachings. The scholars were exposed to videos, case studies, and experiential learning opportunities where they could observe best practices. The faculty and clinicians who worked with the scholars streamlined information that can be complex and overwhelming into a simplified framework—the 4Ms. In the 2020 cohort, four of the six scholars had the 4Ms focus on their capstone projects, and, in 2021, five of the eight scholars focused on the 4Ms.

CIRCAA enhanced the interprofessional experiences for the scholars. Prior to CIRCAA, they described their experience as “lacking”—lacking the chance to experience the reality of professionals from multiple disciplines having opportunities to work with each other. Visiting practice sites and seeing interprofessional teams interact “on-the-job” also enhanced their learning. The scholars felt like they were partnering with the faculty and clinicians teaching them instead of only learning and collaborating with them. Scholars felt exhilarated from the interprofessional training they received in CIRCAA because it provided the “big picture” and was a blending of the interprofessional perspective from both clinicians and faculty researchers. One scholar stated:


*“The health care climate is changing so rapidly that it is important to remain open minded and learn from each other”.*


The scholars expressed the importance of the clinicians observing what is going on in a faculty person’s classroom. At the same time, what is being learned in the classroom needs to be practiced in the community:


*“Person-centered care needs to be practiced. We learned to feel comfortable using the term ‘what matters’ instead of values and preferences. It is important to just plain ask ‘what matters to you?’”*


#### 3.3.2. Innovating

CIRCAA shifted the way the scholars normally teach students about working with older adults and provided new ways to deliver challenging content. Using the 4Ms framework was innovating because it provided room to learn about what matters to the client. It allowed the scholars to escape from their known and comfortable silos:


*“I thought starting off the program using the 4Ms and emphasizing that in the beginning was a great framework to do interprofessional education because there are these four aspects, they involve all of us and it reframes and refocuses our goal”.*


CIRCAA was innovating for the scholars because the interprofessional learning and the 4Ms framework strengthened their practice.

#### 3.3.3. Leveraging

It is important to build on the positive experiences the CIRCAA scholars had and use their testimonials to leverage the sustainability of the program. The scholars expressed a sincere interest in seeing CIRCAA sustained over time and discussed how the program was a good choice over participating in a traditional certificate program. Though often overwhelmed with the amount of content provided, scholars appreciated the sharing of resources. As stated by one of the scholars: “I learned a lot of stuff!”

#### 3.3.4. Challenging

The CIRCAA scholars learned to push past their comfort zones. The challenges noted by the scholars during their CIRCAA training included receiving a plethora of resources—they suggested less volume and content. It was also suggested to use a flipped classroom approach where the scholars are provided readings ahead of time and have more discussions and experiential learning opportunities to apply what they were reading. The scholars expressed that it is important to teach students how to motivate older adults from an interdisciplinary perspective so that changes and improvements can occur. Participants wanted to learn more, but they were unsure of what to do or how to do it until they participated as a CIRCAA scholar. Older adults and professionals need to work together to make a difference in the lives of older adults. Life’s complexities often created some challenges with completing the CIRCAA program. As one scholar stated: “This is my last hurrah. I wanted to be challenged and you have certainly done that!”

Generally, professionals learn to practice within their disciplines. The concept of siloing emerged from the data and is being defined as a limiting perspective based on one’s discipline. One scholar stated that exposing yourself to other disciplines’ perspectives is important:


*“You don’t realize how limited and limiting [your perspective] is until you perhaps, for various reasons, and this was an opportunity here, are exposed to other disciplines and see how the state of healthcare is what it is because of the disconnect in the relationships that need to be occurring for better outcomes to take place”.*


### 3.4. Understanding and Implementing an Interprofessional Age-Friendly Framework

In addition to discussing the strengths and challenges of CIRCAA, another core focus group question pertained to the 4Ms and the value of the 4Ms framework. Two themes emerged from this discussion: impacting and cultivating.

#### 3.4.1. Impacting

Scholars completed CIRCAA with a perspective of receiving something of “tremendous value”. They expressed an understanding of the importance of interprofessional perspectives. One participant stated that an interprofessional perspective “gives me street cred”. Another scholar expressed how impactful CIRCAA was by stating:


*“We came in with an expertise in a quadrant. Right from the first case I felt like, OK, I’m going to hone in on what matters to this person. But we had a PT that was raising issues about the motor and movement and mobility. We had a pharmacist that was bringing in all the issues around medication. And then we had people that were working either in Alzheimer’s or nursing with cognition. It was just so neat for us all to come and share those perspectives together, because I think what happened was we all got an appreciation for the contributions that each of those areas have because of our own comfort level and expertise in those areas”.*


CIRCAA gave the scholars an opportunity to be exposed to interprofessional practice by seeing the impact the interprofessional team of faculty and practitioners had on each other during each training: “I am gaining a lot from watching the faculty and clinicians [instructors] work interprofessionally”. The scholars worked to apply what they were learning during their time in CIRCAA instead of waiting until their scholar experience was complete:


*“I also want to emphasize something that was really big that I’m implementing more is the social determinants of health, implementing some of the questionnaires that we had talked about, like food security into my practice that has a lot to do with diabetes, things that I hadn’t thought to include but now feel are vital to include”.*


The 4Ms framework is a model that was easy to grasp and emphasized the importance of person-centered care and integrating what matters to their clients in their daily practice. One scholar emphasized the importance of escaping the silos of your own discipline by stating:


*“When I think back to about October, I was in K’s group and she was coming all at it from the pharmacist perspective and we were saying, ‘now wait, there’s other things to consider.’ We were like, ‘we need a social worker. We need that information for this case.’ So recruit social workers!”*


#### 3.4.2. Cultivating

The scholars cultivated relationships with each other as well as with the instructors “making friends while learning”. One scholar was at the end of her career, and her experience in CIRCAA gave her a chance to cultivate future opportunities, even after retiring.


*“I know OT really well, but not OT with a geriatric population. I really hope I can contribute even in some guest speaking lectures and things like that, because I’ve learned so much”.*


As a clinician, she had years of experience but not with an older population. She wants to give back to others after her experience as a CIRCAA scholar in a different way than she had anticipated.

## 4. Discussion

Our quantitative findings revealed significant gains in scholars’ perceptions of interprofessional care as well as knowledge about critical components of age-friendly care. Quantitative results from this study showed a concrete improvement in the team skills necessary for quality in the person-to-person interactions occurring in an interprofessional context. Two major themes emerged from the qualitative analysis, both individual preparations to become an interprofessional team member and a comprehensive understanding of the 4Ms framework for advancing age-friendly health systems. Scholars valued the opportunity to share perspectives and enjoyed participating in a program that allowed them to become an interprofessional cohort. The scholars discussed innovative ways to cultivate an atmosphere where multi-disciplinary perspectives are allowed to be normalized. They also expressed the importance of having a stronger skill set after the program to better make interdisciplinary work happen where it is not happening. Our qualitative results indicate that the competency-based curriculum developed for CIRCAA ignited a spark that subsumed the 4Ms individually and conceptually. There was an increased awareness of the SDOH operating within an aging individual’s lifespace and an acknowledgement of the various contexts at work over the course of life.

The significant increase in interprofessional interactions and values and the increase in knowledge of the 4Ms framework, age-friendly health systems, person-centered care, and interprofessional roles in aging care underpin the themes that emerged in creating readiness and understanding and implementing age-friendly interprofessionalism. These themes enrich the ongoing revision process of the CIRCAA program and encourage the integrated delivery of content, with a dedicated focus on the use of the 4Ms as a universally understood interprofessional framework. A partnership between program faculty and scholars through modeling an interprofessional practice in the delivery of teachings, the application of knowledge in practice, and observation in community settings within the 4Ms framework fosters accomplishment of this mutual goal. This frame of reference allows a common ground to develop interprofessional person-centered goals for aging adults and fosters escape from siloed practice. This does not come without challenges, as evidenced by scholars in health settings without readily accessible interprofessional connections. Debriefing after each CIRCAA learning experience creates the space for scholars and program faculty to discuss gaps, create solutions, and reveal what is lacking from an interprofessional perspective in meeting the challenges of addressing what matters most for aging clients and caregivers. These opportunities emphasize and reinforce the competencies scholars need to cultivate and incorporate age-friendly interprofessionalism into their practices.

The quantitative and qualitative results reinforce the importance and usefulness of PDSA in creating awareness and implementation of age-friendly practice in ongoing revisions of the CIRCAA program. The greatest knowledge gains pertained to the 4Ms (90%) and age-friendly, person-centered care (80%), while collaborating with people outside the scholars’ disciplines was the most diminished self-efficacy improvement (65%). Program faculty continuously evaluate the CIRCAA program after each session and each cohort. This process directly informs both what needs to be performed before the next session and what changes are necessary for subsequent cohort years. Learning from each other in this way helps foster the strong relationships that result from CIRCAA participation. Many of the faculty remain engaged with the scholars, both because of ongoing capstone projects and because of a mutual passion for geriatrics education. Currently, 8 of the 25 plenary members are former scholars.

While we find that both quantitative and qualitative analyses are helpful in planning future CIRCAA cohorts through PDSA, there are some limitations, which include those imposed by COVID and the small sample size. The first cohort participated in person until the pandemic began in March, 2020, and the second cohort participated virtually. The first cohort attended two in-person retreats and had opportunities to network within the cohort and with the faculty that were not possible after the pandemic arrived. Session length was also shortened from five hours of teaching to three hours, with extended preparation for the virtual sessions. However, monthly session evaluations were similarly positive when comparing the two cohorts. We also noticed benefits of pivoting to a virtual format, which provided opportunities for scholars to participate from other states and while traveling.

## 5. Conclusions

The focus of this paper is an evaluation of individuals participating in CIRCAA, and how they applied this training to their practice. We also show how interdisciplinary curriculum in geriatrics can be adapted to successfully incorporate age-friendly principles in clinical practice and share them as a model for creating interprofessional learning experiences for a variety of disciplines. The findings suggest that programs such as CIRCAA have the potential to improve older adults’ health by addressing the SDOH, advancing age-friendly and patient-centered care, and promoting an interprofessional model of evidence-based practice. Whereas the previous FDP focused on geriatrics, CIRCAA states “the goal of this program is to create age-friendly practitioners who are knowledgeable in person-centered care”. These are significant changes that point CIRCAA toward age-friendly and patient-centered holistic care and beyond the FDP’s emphasis on geriatrics education and curriculum design.

Many barriers exist when implementing the 4Ms in practice, including who is missing and what is missing; who should be part of the team and how do we get them on the team. CIRCAA provides a streamlined framework for modeling how an interprofessional team works and what this team looks like in order to provide true person-centered care. By refocusing and integrating 4Ms content into the CIRCAA curriculum, which had been informed by geriatric and gerontology competencies, scholars were allowed to escape their silos in order to experience a true person-centered care model. The next steps are to address these barriers and provide incentives to develop healthcare settings such as the Program for All Inclusive Care for the Elderly (PACE) so that professionals can more easily escape these silos in practice through the application of the 4Ms framework.

The National Academy of Medicine’s 2021 Vital Directions for Health and Health Care: Priorities for 2021 initiative [17] include the “creation of an adequately prepared workforce” as a priority across settings through a coordinated interdisciplinary approach, which included continuing education for professionals in the workforce. This terminology suggests that we are not yet an age-friendly workforce, having little impact since the National Academy of Medicine’s 2008 list of priorities was published [6]. We have a journey ahead of us that includes a systematic adoption of the 4Ms in our existing healthcare system. Our CIRCAA faculty responded to this by aligning both discipline-specific geriatric competencies and Interprofessional Education Collaborative competencies to the 4Ms framework within our curriculum. This easily replicated model is just one small but necessary step in the overall goal to create age-friendly ecosystems [18] comprised of health and public health systems, governments, and academic institutions.

Moving forward, to further evaluate the impact of CIRCAA or other geriatrics faculty and clinician programs, it would be worthwhile to follow-up with scholars after their participation. For example, it would be good to assess the inclusion of the 4Ms in participating scholars’ practice and/or teaching in the months and years following CIRCAA completion. Tracking the number and types of practice and teaching sites that these projects touch would also inform the program’s impact. Our curriculum highlighted the utility of making connections between patients and the wealth of resources available in the community. As a consequence, community resources were an aspect woven throughout the mapping of competencies for CIRCAA. Investigations into how these connections are made and sustained as a way to promote age-friendly health care are warranted. CIRCAA emphasized the scholars’ roles as neutral brokers in this process and aimed to make them more adept at locating and referring patients to community services. Essentially, further studies into how geriatrics education programs for faculty and clinicians can contribute to the age-friendly ecosystem are needed. Establishing linkages among healthcare systems and community-based organizations is an equally important aspect of the age-friendly initiative [19]. Replication of CIRCAA or similar models will ultimately lead to a greater infusion of geriatrics, interprofessional collaboration, and incorporation of the 4Ms framework into current practice and educational training.

## Figures and Tables

**Table 1 ijerph-19-05258-t001:** Knowledge Gains.

	Strongly Disagree	Disagree	Neither Agree nor Disagree	Agree	Strongly Agree
I am more knowledgeable about age-friendly health systems.	-	-	-	5 (25%)	15 (75%)
I am more confident I can be an advocate in my workplace for age-friendly, person-centered practices.	-	-	-	5 (25%)	15 (75%)
I am more knowledgeable about the 4Ms framework.	-	-	-	2 (10%)	18 (90%)
I am better able to apply the 4Ms framework in my daily practice.	-	-	-	5 (25%)	15 (75%)
I intend to share what I learned with others in my workplace.	-	-	-	5 (25%)	15 (75%)
I am more confident collaborating with people outside my discipline.	-	-	1 (5%)	6 (30%)	13 (65%)
I am more knowledgeable about age-friendly, person-centered care.	-	-	-	4 (20%)	16 (80%)
I am better able to describe how person-centered care relates to the 4Ms.	-	-	1 (5%)	4 (20%)	15 (75%)
I am more knowledgeable of the roles of other disciplines when caring for complex older adults.	-	-	1 (5%)	4 (20%)	15 (75%)

Note. *N* = 20.

## Data Availability

All data are stored in REDCap, a secure web-based data platform that is HIPPA-compliant.

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
