# Peer review of "Creating Interprofessional Readiness to Advance Age-Friendly U.S. Healthcare"

_ijerph, 2022, doi:10.3390/ijerph19095258_

Round 1

Reviewer 1 Report

Thank you for giving me an opportunity to review this manuscript, creating interprofessional readiness to advance age-friendly healthcare. I think this study has much potential to contribute to the literature as well as to the field. Nonetheless, there are several points that the authors could consider when they revise the manuscript. 

  1. The authors could indicate the location of the study (e.g., in the U.S.) in the title. 
  2. The main strength of this study is the mixed method. I think the author can emphasize this either in the title, the abstract, or the method section. 
  3. Is there any reference(s) for 4Ms (p.2)?
  4. Instead of questions for grounded theory (p.4, last paragraph), I would suggest the authors include core questions that they used for their focus group interviews. Also, there needs reference for grounded theory.
  5. What is the n for Table 1? 
  6. A very small number of survey participants (quantitative method) could be a limitation of this study, which should be acknowledged in the paper. 
  7. The limitations and suggestions for future studies should be described in the discussion/conclusion section. 

Author Response

  1. The authors could indicate the location of the study (e.g., in the U.S.) in the title. Added

  1. The main strength of this study is the mixed method. I think the author can emphasize this either in the title, the abstract, or the method section. Added to the abstract, line 24, and expanded on lines 377-79.
  2. Is there any reference(s) for 4Ms (p.2)? Added Reference 1, line 52.
  3. Instead of questions for grounded theory (p.4, last paragraph), I would suggest the authors include core questions that they used for their focus group interviews. Added Appendix C, referenced on line 187. Also, there needs reference for grounded theory. Added reference [16] on line 189
  4. What is the n for Table 1? Added as a note to the table on line 214
  5. A very small number of survey participants (quantitative method) could be a limitation of this study, which should be acknowledged in the paper. Added reference on limitations line 392
  6. The limitations and suggestions for future studies should be described in the discussion/conclusion section. Limitations in discussion section 390-400 and new suggestions for future studies added to Conclusion section 439 -454.

Reviewer 2 Report

The Age-Friendly Health Systems Initiative  was created to meet the challenges provided by a growing population of older adults, whose care needs are diverse and complex. The goal of the resulting program, Creating Interprofessional Readiness for Complex and Aging Adults (CIRCAA), was to advance age-friendly practice using evidence-based strategies to support wellness and improve health outcomes while also addressing the social determinants of health.  The purpose of the paper  is to detail the creation of CIRCAA program and share insights from  those who have completed the program.

The manuscript is well written and has a clear friendly structure (Introduction, Methods, Results,  Discussion and Conclusions). The subject is  interesting and useful,  as the paper deals with  the program that would help to advance age-friendly care. The background is transparent and informative. The methods, both quantitative and qualitative, are clearly and thoroughly  described. The discussion is sufficient and   is supplemented with limitation section.  The paper has 18 adequate references.  The text is complemented by one table and  Appendix A and B.

 However, there are a few points that can be improved / completed:

  1. Did you compare the results (quantitative and qualitative) between the two cohorts of participants: personal training and virtual training? What differences did you find? What were the most significant benefits and challenges reported by the participants of these two different training modalities? It would be worthwhile to broaden the results on this issue.
  2. Could you explain a bit more precisely limitations imposed by COVID? There were no other limitations?
  3. In subsection 1.2 Program Development abbreviations : ‘BSN and NP’ should be explained (line 92).
  4. As there are many abbreviations in the article that make it difficult to read the manuscript fluently, I propose to re-present them with an appropriate explanation at the end of the main text.

Author Response

  1. Did you compare the results (quantitative and qualitative) between the two cohorts of participants: personal training and virtual training? What differences did you find? What were the most significant benefits and challenges reported by the participants of these two different training modalities? It would be worthwhile to broaden the results on this issue. This would be reasonable if Year-1 all delivered in-person, with Year-2 being all virtual. But Year-1 was virtual for the last 4 months. The comparison of qualitative responses wouldn't be rigorous enough, and therefore wouldn't inform questions about the utility of distance education as a mode of delivery for advancing age-friendly healthcare. Another issue is that scholars who signed up for in person might be different than those who would choose to participate virtually—in both years, scholars elected to join knowing the format available.
  2. Could you explain a bit more precisely limitations imposed by COVID? There were no other limitations? We added more detailed limitations in the paragraph 390-400.
  3. In subsection 1.2 Program Development abbreviations: ‘BSN and NP’ should be explained (line 92). Changes made
  4. As there are many abbreviations in the article that make it difficult to read the manuscript fluently, I propose to re-present them with an appropriate explanation at the end of the main text. We eliminated many of the abbreviations and only kept abbreviations that were fully described and used throughout, CIRCAA, 4Ms, SDOH, and IPEC
